# Associations of Anti-Aquaporin 5 Autoantibodies with Serologic and Histopathological Features of Sjögren’s Syndrome

**DOI:** 10.3390/jcm8111863

**Published:** 2019-11-03

**Authors:** Sumin Jeon, Jennifer Lee, Sung-Hwan Park, Hyun-Duck Kim, Youngnim Choi

**Affiliations:** 1Department of Immunology and Molecular Microbiology, School of Dentistry and Dental Research Institute, Seoul National University, 101 Daehak-ro, Seoul 03080, Korea; pinkngold@naver.com; 2Division of Rheumatology, Internal Medicine, Seoul St. Mary’s Hospital, The Catholic University of Korea, 222, Banpo-daero, Seoul 06591, Korea; poohish@naver.com (J.L.); rapark@catholic.ac.kr (S.-H.P.); 3Department of Preventive and Social Dentistry, School of Dentistry, Seoul National University, 101 Daehak-ro, Seoul 03080, Korea; hyundkim@snu.ac.kr

**Keywords:** Sjogren syndrome, autoantibody, aquaporin 5, epitope, cell-based immunofluorescence cytochemistry, ELISA

## Abstract

Biomarkers to stratify the complex and heterogeneous phenotypes of Sjogren’s syndrome (SS) are needed. We aimed to validate the prevalence of anti-aquaporin 5 (AQP5) IgG in a non-Korean cohort and to optimize the method to screen the anti-AQP5 IgG. The study cohort included 111 primary SS and 43 non-SS Sjögren’s International Collaborative Clinical Alliance (SICCA) controls that were obtained from the Sjögren’s International Collaborative Clinical Alliance registry, in addition to 35 systemic lupus erythematosus (SLE) and 35 rheumatoid arthritis (RA) phenotypes. Anti-AQP5 IgG was screened by cell-based immunofluorescence cytochemistry (CB-IFC) assay in the absence or presence of epitope peptides, as well as by ELISA using the epitope peptides as coated antigens. Anti-AQP5 IgG specific to an E1 epitope was best at discriminating between SS and non-SS, and the two different methods (CB-IFC and ELISA) presented comparable performance in diagnostic accuracy (0.690 vs. 0.707). Notably, the SLE and RA groups had substantially lower levels of anti-AQP5 IgG than the SS group. In addition, the presence of anti-AQP5_E1 IgG was associated with serologic and histopathological features of SS. In conclusion, a similar prevalence of anti-AQP5 IgG was confirmed in a non-Korean cohort. Screening anti-AQP5 autoantibodies may help to form subgroups of SS for targeted therapy.

## 1. Introduction

Sjögren’s syndrome (SS) is a systemic autoimmune disease characterized by a dry mouth, dry eyes, and focal lymphocytic infiltration of the salivary/lacrimal glands [1]. In addition to the histopathologic change in the glands, serologic abnormalities such as autoantibodies to SS-related antigen A (SSA) or SS-related antigen B (SSB), antinuclear antibodies (ANAs), rheumatoid factors (RFs), and hypergammaglobulinemia are often detected in SS patients. The 2002 American–European Consensus Group (AECG) criteria, the most widely used criteria in clinical settings and research, considered symptoms of dryness as well as objective items as key elements of diagnosis [2]. With increased demands to classify patients for enrollment in clinical trials, the 2016 American College of Rheumatology (ACR)/European League Against Rheumatism (EULAR) criteria based only on objective tests were developed, in which the diagnosis of SS depends on the final score of five objective tests with a weighted score [3]. Agreement between the two criteria was high, and the 2016 ACR/EULAR criteria showed higher sensitivity but lower specificity than the 2002 AECG criteria [4,5]. However, mere identification of SS is not good enough. Since SS is a highly heterogeneous disease that manifests both organ-specific and systemic features [6,7], a uniform response is unlikely to occur with most treatments. For a successful clinical trial, the identification of well-defined subpopulations is important. Thus, biomarkers to stratify the complex phenotypes of patients are needed.

Dryness has an overwhelming effect on the quality of life of SS patients. The mechanisms underlying the glandular dysfunction in SS are not fully understood, but potential mechanisms include T-cell-mediated destruction of glands, the presence of anti-muscarinic 3 receptor autoantibodies, modified aquaporin-5 (AQP5) distribution, and the presence of anti-AQP5 autoantibodies [8,9,10,11,12]. AQP5 is predominantly expressed in the apical membranes, but is also expressed in the basolateral membranes of human salivary gland acini [13]. AQP5 is composed of six transmembrane helices connected with three extracellular (A, C, and E) and two intracellular (B and D) loops [13]. We previously reported the presence of anti–AQP5 immunoglobulin G (IgG) in Korean patients with primary SS with a sensitivity of 0.73 and a specificity of 0.68 [11]. We also characterized the epitopes of anti-AQP5 IgG localized to the three extracellular loops [12]. To serve as the biomarker of SS, the validation of anti-AQP5 IgG and its assay method is important.

The aims of this study were to validate the prevalence of anti-AQP5 IgG in a non-Korean cohort and to optimize the method for the detection of anti-AQP5 IgG. Here, we report the presence of anti-AQP5 autoantibodies in a non-Korean cohort and strong associations of the anti-AQP5 autoantibodies with serologic and histopathological features of SS.

## 2. Experimental Section

### 2.1. Serum Samples and Clinical Data

This study was performed in compliance with the Helsinki Declaration after approval from the Institutional Review Board at Seoul National University, School of Dentistry (IRB number: S-D20170004). Due to the fact that all used samples were from tissue banks, a written informed consent for the current study was waivered. For this case-control study, 154 serum samples of Asian females, including 111 primary SS patients and 43 non-SS controls, all of whom had signs and symptoms suggestive of SS, were obtained from Sjögren’s International Collaborative Clinical Alliance (SICCA) registry. The SS samples fulfilled the 2016 ACR/EULAR criteria. In addition, the serum samples of rheumatoid arthritis (RA, *n* = 35) and systemic lupus erythematosus (SLE, *n* = 35) patients without signs and symptoms suggestive of SS were obtained from the Department of Rheumatology, Seoul St. Mary’s Hospital. RA and SLE were chosen as disease controls because they are most prevalent other than SS among the connective tissue diseases, particularly in Korea [14,15]. The demographic and laboratory characteristics of the subject groups are summarized in Table 1. Accompanying clinical and laboratory data were also obtained from the serum providers. Due to the shortage in the available amount of sera, SLE and RA samples were subjected only to screening of anti-AQP5 IgG by ELISA.

### 2.2. Cell-Based Immunofluorescence Cytochemistry (CB-IFC)

Madin-Darby canine kidney (MDCK) cells expressing full length human AQP5 (MDCK-AQP5) were cultured in DMEM with 10% FBS and 1% penicillin and streptomycin in the presence of 2 mg/ml G418 [16]. A mixture of MDCK and MDCK-AQP5 at 1:1 was plated onto collagen-coated coverslips 12 mm in diameter. The cells were stimulated with 0.5 mM cAMP for 24 h, fixed with 4% paraformaldehyde in phosphate-buffered saline (PBS), and then subjected to antigen retrieval by incubation in sodium citrate buffer (10 mM sodium citrate, 0.05% Tween-20, pH = 6) at 105 °C for 20 min. After blocking with 5% bovine serum albumin (BSA) in PBS, the cells were incubated with mouse monoclonal antibodies (a clone D-7) raised against a peptide near the C-terminus of human AQP5 (Santa Cruz Biotechnology, Dallas, TX, USA), along with human serum (1:10 for IgA and 1:100 dilutions for IgG). In the case of anti-AQP5 IgG screening, cells were incubated in parallel with sera preincubated overnight with 10 μg/ml synthetic peptide A, C2, or E1 in RIA buffer (10 mM Tris-HCl, 350 mM NaCl, 1% BSA, 1% Triton X-100, 10% horse serum, pH = 7.6). The sequences of peptides have been previously reported [12]. Subsequently, the cells were stained with Alexa Fluor 488–conjugated rat anti-mouse IgG (Jackson ImmunoResearch, West Grove, PA, USA) and either Alexa Fluor 555–conjugated goat anti-human IgG (Invitrogen, Carlsbad, CA, USA) or Alexa Fluor 594–conjugated rabbit anti-human IgA (Jackson ImmunoResearch, West Grove, PA, USA). After mounting, the cells were examined by confocal microscopy (Carl Zeiss, Oberkochen, Germany). At least 3 areas of AQP5-expressing cells were randomly selected based on staining by the mouse anti-AQP5 antibody and imaged sequentially for staining by either human IgA or IgG. After coding the images, the relative intensities of the anti-AQP5 human Ig signals were blindly determined by decreasing the brightness of the red signal until all signals of the AQP5 stain disappeared (Appendix A). Detail methods have been previously described [16].

### 2.3. Enzyme-Linked Immunosorbent Assay (ELISA)

Ninety-six-well MaxiSorp microtiter plates (Corning, Corning, NY, USA) were precoated with 1 μg/mL avidin in PBS at 4 °C overnight. The wells were washed with washing buffer (0.1% Tween 20 in PBS) and then blocked with Superblock^TM^ blocking buffer (ThermoFisher Scientific, Waltham, MA, USA) at room temperature for 1 h. Subsequently, 0.2 μg biotinylated peptide antigen diluted in 0.1 mL 1% BSA in PBS was added to each well and incubated for 1 h. Sera diluted in 1% BSA in PBS were then incubated in duplicate for 1 h. Sera were also incubated with wells coated with avidin alone in parallel. After washing, bound human Ig was detected with either horseradish peroxidase (HRP)-conjugated goat anti-human IgA (SouthernBiotech, Birmingham, AL, USA) or HRP-conjugated goat anti-human IgG (SouthernBiotech, Birmingham, AL, USA), followed by 3,3’5,5’-tetramethylbenzidine dihydrochloride (Merck, Kenilworth, NJ, USA). Standard curves were generated by directly coating the first two columns with serial dilutions (0–50 ng/mL) of commercial human IgG or IgA. The absorbance was measured at 450 nm. To remove natural antibodies to avidin in human serum [17] and nonspecifically bound antibodies, the concentration of autoantibodies against peptides was calculated using the standard curve and the OD value after subtraction with that obtained from incubation with avidin alone.

### 2.4. Statistical Analysis

Due to the fact that the levels of antibodies did not pass the normality test in all groups, nonparametric analyses were used. Differences in the levels of anti-AQP5 autoantibodies between groups were determined by the Mann–Whitney *U* test or Kruskal–Wallis test followed by a post hoc test with Dunn–Bonferroni correction. The diagnostic powers of anti-AQP5 autoantibodies were analyzed by a receiver operating characteristic (ROC) curve, used to evaluate the c-index producing the area under the curve (AUC). Cut-off values were set to achieve maximal accuracy. The association between the presence of anti-AQP5 autoantibodies and the phenotypic features of SS was explored by two approaches: 1) a contingency table approach with either chi-square or Fisher’s exact test, 2) a nonparametric approach with a Mann–Whitney *U* test to compare the levels of anti-AQP5 autoantibodies as a continuous variable by the presence or absence of SS phenotypic features. The association between the unstimulated whole salivary flow rate (UWSFR) and the presence of anti-AQP5 autoantibodies was explored by the Mann–Whitney *U* test, used to compare UWSFR as a continuous variable by the presence or absence of anti-AQP5 autoantibodies. *p* < 0.05 was considered to indicate a statistically significant difference. All statistics were performed using SPSS software version 23.0 (IBM, Chicago, IL, USA).

## 3. Results

### 3.1. Higher Levels of Anti-AQP5 IgG and IgA Were Detected in SS than in Non-SS by CB-IFC

To determine the prevalence of anti-AQP5 autoantibodies in the non-Korean SS cohort, 154 serum samples of Asian females, including 111 SS patients and 43 non-SS controls, all of whom had signs and symptoms suggestive of SS, were obtained from the SICCA registry. The characteristics of the subjects are summarized in Table 1. The presence of anti-AQP5 IgG was determined by CB-IFC using a 1:100 dilution of sera and MDCK-AQP5 cells. In the absence of epitope peptides, AQP5-stained human IgG was detected in all samples, the signal of which was significantly reduced by preincubation of sera with epitope peptides in the SS group but not in the non-SS group (Figure 1A). By subtracting the signal intensities obtained in the presence of epitope peptides from that obtained in the absence of peptide, the levels of each epitope-specific IgG were determined. The SS group had significantly increased levels of not only anti-AQP5 IgG, but also anti-AQP5_A, anti-AQP5_C2, and anti- AQP5_E1 IgG compared to the corresponding levels in the non-SS group (Figure 1B). The levels of AQP5-staining IgA determined using the 1:10 dilution of sera were also significantly higher in SS than in non-SS (Figure 1C). ROC curve analysis revealed that anti-AQP5_E1 IgG had the greatest power to differentiate SS from non-SS based on the AUC, resulting in a sensitivity of 0.61 and a specificity of 0.77 (Figure 1D and Table 2).

### 3.2. Higher Levels of Anti-AQP5 IgG but Not IgA were Detected in the SS Sera by ELISA

Direct coating of the epitope peptides did not yield much binding of antibodies. Thus, we developed an ELISA to screen anti-AQP5 autoantibodies by utilizing an avidin-biotinylated peptide coating system that helped to maintain the structures of immobilized peptides (Figure 2A). Each epitope peptide (A, C2, E1), or the mixture of the three peptides, was coated, and concentrations of IgG or IgA that bound to the coated peptides were determined by ELISA. To determine the presence of anti-AQP5 autoantibodies in other autoimmune diseases, in addition to the SS and non-SS samples, the samples from SLE (*n* = 35) and RA (*n* = 35) patients were also included in the screening of IgG. The coating of mixed peptides did not result in additive detection of anti-AQP5 autoantibodies. The levels of IgG bound to each antigen were significantly higher in the SS group than in the other groups. In general, the anti-AQP5 IgG was less frequently detected in either SLE or RA than in the non-SS control, and the levels of anti-AQP5_A and anti-AQP5_C IgG determined using 1:100 diluted sera were significantly lower in RA than in non-SS groups (Figure 2B). In contrast to the results obtained by CB-IFC, the levels of the anti-AQP5 IgA determined by ELISA were not significantly different between the SS and non-SS groups (Figure 2C).

To compare the accuracy between ELISA and CB-IFC, ROC curve analysis was performed only for the SICCA registry samples. The highest accuracy was achieved by anti-AQP5_E1 IgG determined using the 1:50 dilution of serum, resulting in a sensitivity of 0.86 and a specificity of 0.56 (Figure 2D and Table 2). In the case of IgA, anti-AQP5_AC2E1 IgA determined using the 1:20 dilution of serum yielded the highest AUC (0.598, 95% confidence interval 0.496–0.700), but this result was not significant (*p* = 0.06). When the SLE and RA samples were included for ROC curve analysis, the highest accuracy was obtained by anti-AQP5_A IgG determined using the 1:100 dilution of serum, resulting in a sensitivity of 0.89 and a specificity of 0.64. However, the cut-off value obtained from the analysis with inclusion of SLE and RA was less effective in the discrimination between SS and non-SS.

### 3.3. Association between Anti-AQP5 Autoantibodies and Disease Criteria for SS

Associations between the anti-AQP5 autoantibodies and disease criteria for SS in the SICCA registry samples were analyzed first by either chi-square or Fisher’s exact testing. Anti-AQP5_E IgG positivity, determined by either CB-IFC or ELISA, was associated with positive serum SSA, RF, and ANA (titer > 1:320); focal lymphocytic sialadenitis (FLS) score ≥1; and ocular staining score ≥3 (Table 3). The anti-AQP5_E1 IgG positivity determined by ELISA had stronger associations, all with *p*-values ≤0.001, than that determined by CB-IFC. Anti-AQP5 IgA and anti-AQP5_C2 IgG determined by CB-IFC and anti–AQP5_AC2E1 IgG determined by ELISA (1:50 dilution) also had significant associations with the aforementioned five disease criteria (Appendix A). Anti-AQP5_E1 IgG positivity had a weak association with positive serum SSB, but no significant association with either oral dryness or Schirmer’s test positivity (Table 3).

The relationship of the levels of anti-AQP5 autoantibodies to each disease criteria for SS was further explored by either Mann–Whitney *U* or Kruskal–Wallis *H* testing, which confirmed the associations presented in Table 3 and Appendix A (Table 4).

Since the anti-AQP5_E IgG positivity determined by ELISA had strong associations with positive SSA and a FLS score ≥1, we explored whether it can replace an item of the 2016 ACR-EULAR criteria for the diagnosis of SS. The anti-AQP5_E IgG positivity with weight two could replace the FLS score ≥1 with just a little decrease in the sensitivity (0.973) and specificity (0.953) (Figure 3).

We also explored whether the anti-AQP5_E IgG is associated with systemic diseases or adverse prognostic markers, including RF, hypocomplementemia, hypergammaglobulinemia, high FLS score, and the presence of germinal center among the SS patients [7]. Only the high FLS score was associated with the high titer of anti-AQP5_E IgG (Table 5).

In our previous study, positive anti-AQP5 IgG or IgA was associated with low UWSFR in SS patients [11]. In the current study, the median UWSFR among subjects with positive anti-AQP5_E IgG determined by CB-IFC was 0.05 ml/minute (range 0–0.53) vs. 0.08 ml/minute (range 0–0.58) among those with negative anti-AQP5_E IgG in the SICCA registry samples, including both SS and non-SS (*p* = 0.022). However, this association was not observed in SS samples alone [0.042 (range 0–0.534) vs. 0.046 (range 0–0.420), *p* = 0.439].

## 4. Discussion

In the present study, the prevalence of anti-AQP5 autoantibodies in a non-Korean SS cohort was investigated using multiple assays: CB-IFC in the absence or presence of epitope peptides and epitope peptide-specific ELISA. In addition, strong associations of anti-AQP5 autoantibodies with the serologic and histopathological features of SS were confirmed.

The anti-AQP5 IgG determined by CB-IFC in the absence of epitope peptides, which was the same assay used in the previous study, resulted in a sensitivity of 0.595 and a specificity of 0.721 using the cut-off value that gives a maximal accuracy. The cut-off value to obtain a sensitivity of 0.73, the sensitivity obtained in the previous study, resulted in a specificity of 0.558. Although other reasons can be speculated, the reduced specificity can be attributed mainly to a difference in control subjects between the two studies. While the control subjects in the previous study were all healthy, the non-SS controls in the current study had symptoms suggestive of SS. Importantly, 42 out of the 43 non-SS subjects had a certain type of sialadenitis in the labial salivary glands, including FLS, nonspecific chronic sialadenitis, and sclerosing chronic sialadenitis. Such inflammation in the salivary glands may contribute to the production of anti-AQP5 autoantibodies. Preincubation of sera with epitope peptides substantially decreased the signals of AQP5-staining IgG in SS samples but not in non-SS control samples. Consequently, the use of either C2 or E1 peptides in CB-IFC assays screening for anti-AQP5 IgG improved the diagnostic accuracy.

Among the multiple assays targeting different antigens and Ig isotypes, AQP5_E1-specific IgG was best at discriminating between SS and non-SS subjects, and the two different methods (CB-IFC and ELISA) presented comparable diagnostic performance (AUC: 0.703 vs. 0.695, accuracy: 0.690 vs. 0.707). CB-IFC and ELISA had different pros and cons. The greatest advantage of CB-IFC was the use of natively folded membrane-expressed AQP5 as a target antigen. In contrast, the peptide antigens used in ELISA were mimotopes of AQP5 origin. Spearman’s rank correlation analysis clearly revealed that CB-IFC and ELISA seemed to detect the anti-AQP5 autoantibodies of a few different epitopes: The levels of anti-AQP5 IgG determined by CB-IFC had significant positive correlations with anti-AQP5 IgA and epitope-specific IgG, all determined by CB-IFC, but with none of those determined by ELISA, and vice versa (Appendix A). While the levels of anti-AQP5 IgA determined by CB-IFC were substantially higher in SS than in non-SS, those of epitope peptide-specific IgA determined by ELISA did not show significant differences between the two groups. Although the reason for this discrepancy is not currently clear, differences in epitopes detected by the two methods may have a role. Compared with ELISA, CB-IFC is labor-intensive, time-consuming, and difficult to quantitate. Furthermore, the reproducibility of CB-IFC depends on diverse factors [18] and was inferior to ELISA in our experience. In addition, the anti-AQP5_E1 IgG positivity determined by ELISA had stronger associations with most disease criteria for SS than that determined by CB-IFC (Table 3). Although the anti-AQP5_A IgG determined by ELISA provided the highest accuracy (0.765) for the diagnosis of SS in total samples, including SLE and RA, we recommend anti-AQP5_E1 IgG ELISA as the gold standard for the diagnosis of SS in a clinical setting where the differentiation of SS and non-SS among the subjects with signs suggestive of SS is important.

The anti-AQP5_E1 IgG determined by ELISA had significant associations with other disease criteria for SS, including SSA, RF, ANA, FLS score ≥1, and ocular staining score ≥3, among the SICCA registry samples according to the results of both the bivariate analysis and nonparametric approach. In particular, the anti-AQP5_E1 IgG test could replace the FLS score ≥1 for the diagnosis of SS with reasonable performance (AUC 1 vs. 0.993). The labial salivary gland biopsy still provides critical information about SS, such as the development of lymphoma or the predominating immune cells. The 5% of the cases in the current study failed to obtain a proper FLS score from the biopsy. Only in such cases, the anti-AQP5_E1 IgG test may be used in supplementary.

The association of the high titer of anti-AQP5_E1 IgG with high FLS score (≥3) in the SS group suggests that local production at the salivary glands contributes to the levels of anti-AQP5 autoantibodies. However, the association of the high titer of anti-AQP5_E1 IgG with systemic diseases or other adverse prognostic markers, including RF, hypocomplementaemia, hypergammaglobulinemia, and the presence of germinal center was not observed.

Positivity with anti-AQP5_E1 IgG by CB-IFC was associated with low UWSFR among the SICCA registry samples, but not among the SS samples alone. These results indicate that the presence of anti-AQP5 autoantibodies was not directly associated with oral dryness in SS patients in the current study. This discrepancy with the previous result may be attributed to diverse variables, such as differences in the cohort, the criteria used for the diagnosis of SS, the prevalence of anti-muscarinic 3 receptor autoantibodies, intrinsic issues with assay reproducibility, and the possibility of differential modulation of the function of AQP5 by the anti-AQP5 autoantibodies.

This study has several limitations. First, the SLE and RA subjects did not go through the full workup for the diagnosis of SS because none of the patients fulfilled the inclusion criteria for SS. Notably, the SLE and RA groups had substantially lower levels of anti-AQP5 IgG than the SS group. In particular, the levels of anti-AQP5 IgG targeting A were even lower than non-SS SICCA, suggesting that anti-AQP5 IgG is associated with salivary gland diseases. Unlike the SICCA registry samples, the SLE and RA samples had no associations between the anti-AQP5 IgG and SSA, ANA, or RF (Appendix A). Although there is a possibility that secondary SS patients without SICCA symptoms were included in the SLE and RA groups, the anti-AQP5 autoantibodies seem to be more prevalent in SS than in other autoimmune diseases. Second, this was a cross-sectional study; therefore, the predictive value of the anti-AQP5 autoantibodies could not be evaluated.

Theander et al. reported that various autoantibodies, such as anti-SSA, anti-SSB, RF, ANA, anti-ribonucleoproteins, or anti-chromatin antibodies, are present for up to 18–20 years before the diagnosis of primary SS [19]. None of autoantibodies other than anti-AQP5 autoantibodies were detected in the non-SS SICCA registry subjects included in the current study. Whether the non-SS subjects with anti-AQP5 IgG have an increased prevalence of SS diagnosis in the future is worth answering.

Collectively, we confirmed the presence of anti-AQP5 autoantibodies in a non-Korean cohort and strong associations of the anti-AQP5 autoantibodies with serologic and histopathological features of SS. AQP5 is also expressed in epidermis, sweat glands, vaginal epithelial cells, airway submucosal glands, lung alveoli, kidney cortex, pancreas, and the central nervous system, as well as in salivary and lacrimal glands, which overlaps with organs involved in the systemic manifestations of SS [20,21,22]. Associations between the presence of anti-AQP5 autoantibodies and the systemic manifestations of SS, in particular dryness-related symptoms and the EULAR SS disease activity index (ESSDAI), need to be clarified in future studies. Screening anti-AQP5 autoantibodies may help to form subgroups of SS for targeted therapy.

## Figures and Tables

**Figure 1 jcm-08-01863-f001:**
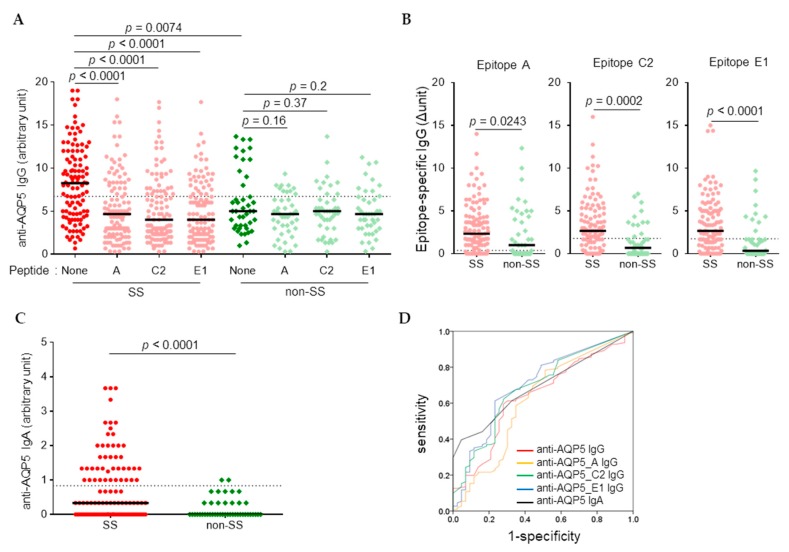
Higher levels of anti-aquaporin 5 (AQP5) IgG and IgA were detected in Sjögren’s syndrome (SS) than in non-SS by cell-based immunofluorescence cytochemistry (CB-IFC). Madin–Darby canine kidney (MDCK) cells overexpressing AQP5 were stained with mouse anti-human AQP5 monoclonal antibodies along with human serum (1:10 for IgA and 1:100 dilutions for IgG) in the absence or presence of epitope peptides, followed by Alexa Fluor 488–conjugated rat anti-mouse IgG and either Alexa Fluor 555–conjugated goat anti-human IgG or Alexa Fluor 594–conjugated rabbit anti-human IgA. (**A**) The intensities of the red signals for anti-AQP5 IgG were expressed by the magnitude of brightness, which was reduced until the staining of AQP5 disappeared. (**B**) The levels of each epitope-specific IgG were determined by subtracting the signal intensities obtained in the presence of epitope peptides from those obtained in the absence of peptide. (**C**) The levels of IgA were expressed by the magnitude of brightness. (**D**) Receiver operating characteristic (ROC) curves of anti-AQP5 IgG and IgA. Dotted lines in graphs A–C present the cut-off values based on the ROC analysis.

**Figure 2 jcm-08-01863-f002:**
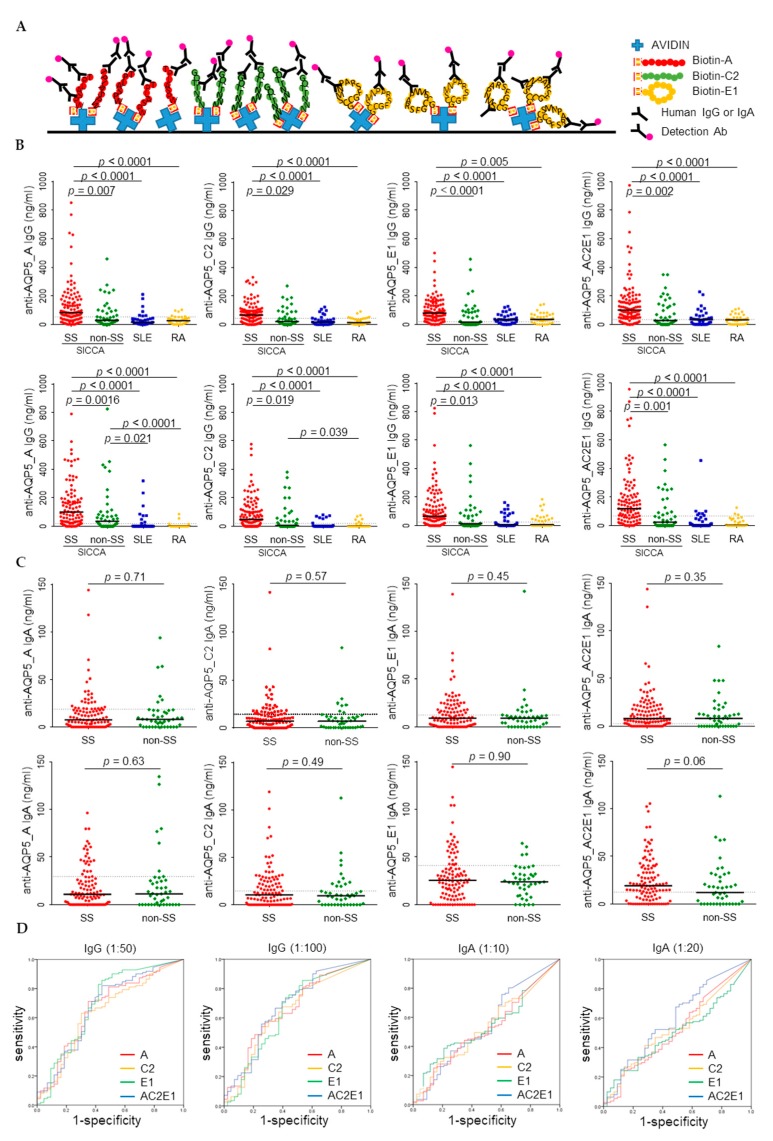
Higher levels of anti-AQP5 IgG were detected in SS than in non-SS, systemic lupus erythematosus (SLE), or rheumatoid arthritis (RA) by enzyme-linked immunosorbent assay (ELISA). (**A**) A diagram of the ELISA strategy used. (**B**) Concentrations of IgG reactive with A, C2, E1, or a mixture of A, C2, and E1 epitope peptides were measured by ELISA using sera diluted at 1:50 (top panel) or 1:100 (bottom panel). (**C**) Concentrations of IgA reactive with A, C2, E1, or a mixture of A, C2, and E1 epitope peptides were measured by ELISA using sera diluted at 1:10 (top panel) or 1:20 (bottom panel). (**D**) ROC curves of anti-AQP5 IgG and IgA. The dotted lines in graphs B–C present the cut-off values based on the ROC analysis.

**Figure 3 jcm-08-01863-f003:**
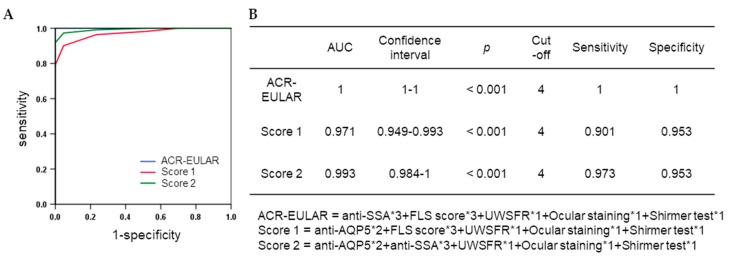
Diagnostic performance of anti-AQP5_E IgG when it replaced either anti-SSA positivity or the FLS score in the 2016 ACR-EULAR criteria. (**A**) ROC curves (**B**) AUC, sensitivity, specificity, and the equations used to obtain each SS criteria in the ROC curves.

**Table 1 jcm-08-01863-t001:** Demographic and laboratory characteristics of the subjects.

		SICCA Registry		Seoul St. Mary’s Hospital	
		SS (*n* = 111)	Non-SS (*n* = 43)	SLE (*n* = 35)	RA (*n* = 35)
Age (years), mean ± SD		50.4 ± 13.6	53.1 ± 14.9	30.5 ± 9.4*	57.0 ± 12.4
anti-SSA+, *n* (%)		102 (91.9)	0 (0)	20 (69.0), 29^†^	ND
anti-SSB+, *n* (%)		61 (55.0)	0 (0)	6 (20.7), 29^†^	ND
RF+, *n* (%)		75 (67.6)	0 (0)	0 (0), 22^†^	27 (77.1)
ANA+, *n* (%)		75 (67.6)	0 (0)	34 (97.1)	14 (45.2), 31^†^
UWSFR ≤0.1 mL/min, *n* (%)		84 (75.7)	15 (34.9)	ND	ND
Labial salivary gland biopsy results	FLS score ≥1, *n* (%)	80 (80), 100^†^	0 (0)	ND	ND
	FLS score 0 << 1, *n* (%)	7 (7), 100^†^	8 (18.6)	ND	ND
	FLS score = 0, *n* (%)	13 (13), 100^†^	35 (81.4)	ND	ND
	N/SCS	13 (13), 100^†^	34 (79.0)	ND	ND
Schirmer’s test ≤5 mm in 5 min, *n* (%)		63 (57.3), 110^†^	11 (25.6)	ND	ND
Ocular staining score ≥3, *n* (%)		105 (95.5), 110^†^	0 (0)	ND	ND

ND: not done; SSA: Sjögren’s syndrome-related antigen A; SSB: Sjögren’s syndrome-related antigen B; RF: rheumatoid factor; ANA: antinuclear antibody; UWSFR: unstimulated whole salivary flow rate; FLS: focal lymphocytic sialadenitis; N/SCS: nonspecific/sclerosing chronic sialadenitis. *Significantly different from the other groups by ANOVA with Bonferroni-adjusted post hoc tests; ^†^ presents the total number with proper data.

**Table 2 jcm-08-01863-t002:** Diagnostic performance of anti-AQP5 autoantibodies in 154 Sjögren’s International Collaborative Clinical Alliance (SICCA) registry samples by receiver operating characteristic (ROC) analysis.

Method	Ab	Dilution of Sera	AUC	Confidence Interval	*p*	Cut-off Value	Sensitivity	Specificity	Accuracy
CB-IFC	anti-AQP5_IgG	1:100	0.638	0.542–0.735	0.008	6.71	0.595	0.721	0.658
	anti-AQP5_A IgG	1:100	0.616	0.510–0.721	0.026	0.38	0.784	0.488	0.636
	anti-AQP5_C2 IgG	1:100	0.688	0.595–0.781	<0.001	1.79	0.622	0.721	0.672
	anti-AQP5_E1 IgG	1:100	0.703	0.609–0.796	<0.001	1.75	0.613	0.767	0.690
	anti-AQP5 IgA	1:10	0.694	0.612–0.777	<0.001	0.83	0.396	0.953	0.675
ELISA	anti-AQP5_A IgG	1:50	0.665	0.565–0.764	0.002	54.1	0.712	0.651	0.681
	anti-AQP5_C2 IgG	1:50	0.640	0.540–0.740	0.007	44.3	0.631	0.698	0.664
	anti-AQP5_E1 IgG	1:50	0.695	0.593–0.798	<0.001	19.3	0.856	0.558	0.707
	anti-AQP5_ACE IgG	1:50	0.671	0.570–0.772	0.001	34.9	0.820	0.558	0.689
	anti-AQP5_A IgG	1:100	0.664	0.567–0.762	0.002	20.3	0.811	0.465	0.638
	anti-AQP5_C2 IgG	1:100	0.645	0.544–0.746	0.005	21.6	0.676	0.605	0.640
	anti-AQP5_E1 IgG	1:100	0.649	0.544–0.753	0.004	26.0	0.739	0.581	0.660
	anti-AQP5_ACE IgG	1:100	0.682	0.583–0.781	<0.001	66.0	0.667	0.651	0.659

**Table 3 jcm-08-01863-t003:** Bivariate analysis exploring the presence of anti-AQP5_E1 IgG by disease criteria for Sjögren’s syndrome in 154 SICCA registry samples.

Disease Criteria for Sjogren’s Syndrome		*n*	Anti-AQP5_E1 IgG by CB-IFC			Anti-AQP5_E1 IgG by ELISA		
			≥1.75 (au)	<1.75 (au)	*p* ^*^	≥19.3 (ng/mL)	<19.3 (ng/mL)	*p* ^*^
anti-SSA	Positive	102	**62 (60.8%)**	**40 (39.2%)**	**0.001**	**84 (82.4%)**	**18 (17.6%)**	**<0.001**
	Negative	52	**17 (32.7%)**	**35 (67.3%)**		**27 (51.9%)**	**25 (48.1%)**	
anti-SSB	Positive	61	35 (57.4%)	26 (42.6%)	0.222	**50 (82.0%)**	**11 (18.0%)**	**0.027**
	Negative	93	44 (47.3%)	49 (52.7%)		**61 (65.6%)**	**32 (34.4%)**	
RF	Positive	75	**45 (60.0%)**	**30 (40.0%)**	**0.035**	**63 (84.0%)**	**12 (16.0%)**	**0.001**
	Negative	79	**34 (43.0%)**	**45 (57.0%)**		**48 (60.8%)**	**31 (39.2%)**	
ANA	Positive	75	**46 (61.3%)**	**29 (38.7%)**	**0.015**	**67 (89.3%)**	**8 (10.7%)**	**<0.001**
	Negative	79	**33 (41.8%)**	**46 (58.2%)**		**44 (55.7%)**	**35 (44.3%)**	
UWSFR	≤0.1 ml/min	99	56 (56.6%)	43 (43.4%)	0.079	74 (74.7%)	25 (25.3%)	0.322
	>0.1 ml/min	55	23 (41.8%)	32 (58.2%)		37 (67.3%)	18 (32.7%)	
FLS score	≥1	80	**50 (62.5%)**	**30 (37.5%)**	**0.006**	**70 (87.5%)**	**10 (12.5%)**	**<0.001**
	0 << 1	15	**7 (46.7%)**	**8 (53.3%)**		**10 (66.7%)**	**5 (33.3%)**	
	=0	48	**16 (33.3%)**	**32 (66.7%)**		**24 (50.0%)**	**24 (50.0%)**	
Schirmer’s test	≤5 mm in 5 min	74	36 (48.6%)	38 (51.4%)	0.618	58 (78.4%)	16 (21.6%)	0.391
	>5 mm in 5 min	79	39 (49.4%)	40 (50.6%)		55 (69.6%)	24 (30.4%)	
Ocular staining score	≥3	105	**66 (62.9%)**	**39 (37.1%)**	**<0.001**	**86 (81.9%)**	**19 (18.1%)**	**<0.001**
	<3	49	**13 (26.5%)**	**36 (73.5%)**		**25 (51.0%)**	**24 (49.0%)**	

SSA: Sjögren’s syndrome-related antigen A; SSB: Sjögren’s syndrome-related antigen B; RF: rheumatoid factor; ANA: antinuclear antibody; UWSFR: unstimulated whole salivary flow rate; FLS: focal lymphocytic sialadenitis; au: arbitrary unit. *By either chi-square or Fisher’s exact test. Bold denotes statistical significance at *p* < 0.05.

**Table 4 jcm-08-01863-t004:** Nonparametric approach exploring the presence of anti-AQP5 autoantibodies by disease criteria for Sjögren’s syndrome in 154 SICCA registry samples.

Disease Criteria for Sjogren’s Syndrome		IgA by CB-IFC (au)	IgG by CB-IFC (au)				IgG by ELISA (ng/ml)			
		AQP5	A	C2	E1	AQP5	A	C2	E1	AC2E1
anti-SSA	Positive	**0.3 (3.7)**	2.3 (16.0)	**2.4 (16.0)**	**2.5 (15.0)**	**8.2 (18.3)**	**67 (852)**	**53 (332)**	**66 (500)**	**85 (975)**
	Negative	**0 (3.7)**	1.5 (12.3)	**1.0 (10.3)**	**1.0 (9.7)**	**5.3 (12.7)**	**39 (458)**	**31 (271)**	**35 (457)**	**46 (349)**
	*P* ^†^	**< 0.001**	0.099	**0.008**	**0.002**	**0.019**	**0.005**	**0.016**	**0.002**	**0.004**
anti-SSB	Positive	0.3 (3.7)	2.3 (14.0)	2.3 (12.8)	2.3 (14.3)	7.3 (18.3)	74 (852)	52 (332)	78 (500)	86 (975)
	Negative	0.3 (3.7)	1.8 (12.3)	1.7 (16.0)	1.7 (15.0)	6.8 (18.0)	55 (459)	41 (298)	52 (457)	64 (351)
	*P* ^†^	0.183	0.356	0.243	0.406	0.688	0.253	0.635	0.052	0.208
RF	Positive	0.3 (3.7)	2.3 (14.0)	**2.4 (16.0)**	2.7 (14.3)	8.3 (18.3)	**65 (852)**	45 (332)	**67 (500)**	**71 (975)**
	Negative	0.0 (3.7)	1.7 (12.3)	**1.7 (9.7)**	1.5 (15.0)	6.3 (17.0)	**43 (768)**	30 (309)	**46 (457)**	**42 (787)**
	*P* ^†^	0.099	0.155	**0.035**	0.080	0.247	**0.007**	0.254	**0.003**	**0.002**
ANA	Positive	**0.3 (3.7)**	2.0 (14.0)	**2.5 (12.8)**	2.3 (15.0)	8.3 (18.3)	54 (852)	**44 (332)**	**61 (500)**	**66 (975)**
	Negative	**0.0 (3.7)**	1.7 (12.3)	**1.3 (16.0)**	1.5 (14.4)	6.0 (18.0)	43 (628)	**25 (301)**	**34 (457)**	**46 (546)**
	*P* ^†^	**< 0.001**	0.157	**0.024**	0.055	0.409	0.093	**0.004**	**< 0.001**	**0.012**
UWSFR	≤ 0.1 ml/min	0.3 (3.7)	1.8 (14.0)	2.3 (16.0)	2.3 (15.0)	6.8 (18.3)	66 (852)	47 (332)	66 (500)	82 (975)
	> 0.1 ml/min	0.0 (3.7)	1.8 (10.0)	1.7 (10.3)	1.7 (14.4)	8.0 (16.7)	91 (768)	66 (309)	64 (457)	91 (787)
	*P* ^†^	0.327	0.944	0.329	0.259	0.420	0.381	0.240	0.992	0.166
FLS score	≥ 1	**0.3 (3.7)**	**2.1 (19.3)**	**2.5 (21.4)**	**2.5 (20.7)**	8.0 (18.3)	**84 (852)**	**69 (332)**	**88 (500)**	**102 (975)**
	0 << 1	**0.3 (3.7)**	**0.0 (11.3)**	**1.0 (19.0)**	**1.3 (15.7)**	6.0 (18.0)	**101 (768)**	**31 (309)**	**37 (457)**	**129 (787)**
	= 0	**0.0 (3.3)**	**1.7 (16.0)**	**1.2 (9.2)**	**1.0 (13.7)**	6.0 (11.7)	**38 (279)**	**23 (271)**	**25 (383)**	**30 (349)**
	*P* ^‡^	**0.010**	**0.017**	**0.016**	**0.045**	0.295	**0.010**	**0.031**	**< 0.001**	**0.001**
Schirmer’s test	< 5mm in 5min	0.3 (3.3)	1.8 (19.3)	2.3 (15.8)	1.9 (19.0)	7.5 (18.3)	78 (852)	53 (332)	80 (500)	87 (975)
	≥ 5mm in 5min	0 (3.7)	1.8 (15.7)	1.7 (24.7)	2 (20.7)	6.3 (18.0)	65 (768)	44 (309)	61 (457)	87 (787)
	*P* ^†^	0.388	0.652	0.498	0.894	0.975	0.756	0.942	0.170	0.660
Ocular	≥ 3	**0.3 (3.7)**	**2.6 (14.0)**	**2.7 (16.0)**	**3.0 (15.0)**	**8.3 (18.3)**	**82 (852)**	**67 (332)**	**80 (500)**	**101 (975)**
staining score	< 3	**0.0 (2.7)**	**1.0 (12.3)**	**1.0 (7.0)**	**0.3 (9.7)**	**5.0 (13.0)**	**48 (640)**	**33 (271)**	**22 (457)**	**46 (647)**
	*P* ^†^	**< 0.001**	**0.017**	**< 0.001**	**< 0.001**	**0.007**	**0.041**	**0.049**	**0.001**	**0.013**

The levels of anti-AQP5 autoantibodies are expressed as median (range). SSA: Sjögren’s syndrome-related antigen A; SSB: Sjögren’s syndrome-related antigen B; RF: Rheumatoid factor; ANA: anti-nuclear antibody; UWSFR: Unstimulated whole salivary flow rate; FLS: Focal-lymphocytic sialadenitis; au: arbitrary unit. ^†^ By Mann-Whitney U test. ^‡^ By Kruskal-Wallis H test. Bold denotes statistical significance at *p* < 0.05.

**Table 5 jcm-08-01863-t005:** Bivariate analysis exploring the presence of anti-AQP5_E1 IgG by adverse prognostic markers in 111 SS samples.

Adverse Prognostic Markers			Anti-AQP5_E1 IgG by ELISA					
			Positivity			High Titer (> median)		
		n	≥19.3 (ng/mL)	<19.3 (ng/mL)	*p* ^*^	>79.8 (ng/mL)	≤79.8 (ng/mL)	*p* ^*^
FLS	≥3	41	35 (85.4%)	6 (14.6%)	0.932	**27 (65.9%)**	**14 (34.1%)**	**0.013**
	<3	59	50 (84.7%)	9 (15.3%)		**24 (40.7%)**	**35 (59.3%)**	
GC	Positive	8	6 (75.0%)	2 (25.0%)	0.330	4 (50.0%)	4 (50.0%)	0.876
	Negative	87	76 (87.4%)	11 (12.6%)		46 (52.9%)	41 (47.1%)	
RF	Positive	75	63 (84.0%)	12 (16.0%)	0.652	40 (53.3%)	35 (46.7%)	0.250
	Negative	36	29 (80.6%)	7 (19.4%)		15 (41.7%)	21 (58.3%)	
IgG	>1.445 mg/dL	79	66 (83.5%)	13 (16.5%)	0.771	40 (50.6%)	39 (49.4%)	0.720
	≤1.445 mg/dL	32	26 (72.2%)	6 (18.8%)		15 (46.9%)	17 (53.1%)	
C3	<67 mg/dL	6	5 (83.3%)	1 (16.7%)	0.976	3 (50%)	3 (50%)	0.982
	≥67 mg/dL	105	87 (82.9%)	18 (17.1%)		52 (49.5%)	53 (50.5%)	
C4	<16 mg/dL	21	19 (90.5%)	2 (9.5%)	0.305	11 (52.4%)	10 (47.6%)	0.773
	≥16 mg/dL	90	73 (81.1%)	17 (18.9%)		44 (48.9%)	46 (51.1%)	
Salivary gland enlargement	Positive	12	6 (50%)	6 (50%)	0.974	6 (50.0%)	6 (50.0%)	0.974
	Negative	99	49 (49.5%)	50 (50.5%)		49 (49.5%)	50 (50.5%)	
Systemic disease	Positive	20	16 (80.0%)	4 (20%)	0.705	9 (45.0%)	11 (55.0%)	0.653
	Negative	91	76 (83.5%)	15 (16.5%)		46 (50.5%)	45 (49.5%)	

FLS: focal lymphocytic sialadenitis; GC: germinal center; RF: rheumatoid factor. * By either chi-square or Fisher’s exact test. Bold denotes statistical significance at *p* < 0.05.

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
