# Peer review of "Associations of Anti-Aquaporin 5 Autoantibodies with Serologic and Histopathological Features of Sjögren’s Syndrome"

_jcm, 2019, doi:10.3390/jcm8111863_

Round 1

Reviewer 1 Report

Thank you for the detailed reply to my comments. Your manuscript has improved significantly. You will find further comments below. 

How can you know that the RA/SLE patients did not fulfill the inclusion criteria for SjS classification? Were sicca symptoms systematically assessed? Even if they dont fulfill the entry criteria there maybe patients without symtpomatic sicca; I recommend to re-phrase the limitation.

Why do you think that AQP5 stratifies for targeted therapies? From my perspective this is a very bold statement, and I do not see a strong scientific proof of this. 

Line 32: We do not know whether the infiltration causes dryness, autoantibodies might be involved and other things we dont know jet (as you write later in the text). 

The association to hypergammaglobulinaemia might bias associations to other disease characteristics and immun-complexe manifestations. I would add this to the limitations. 

Author Response

We appreciate your time and helpful comments.

How can you know that the RA/SLE patients did not fulfill the inclusion criteria for SjS classification? Were sicca symptoms systematically assessed? Even if they don’t fulfill the entry criteria there maybe patients without symtpomatic sicca; I recommend to re-phrase the limitation.

→ The limitation regarding the RA/SLE patients has been re-written as following (line 317-325): First, the SLE and RA subjects did not go through the full workup for the diagnosis of SS because none of the patients fulfilled the inclusion criteria for SS. Notably, the SLE and RA groups had substantially lower levels of anti-AQP5 IgG than the SS group. In particular, the levels of anti-AQP5 IgG targeting A were even lower than non-SS SICCA, suggesting that anti-AQP5 IgG is associated with salivary gland diseases. Unlike the SICCA registry samples, the SLE and RA samples had no associations between the anti-AQP5 IgG and SSA, ANA, or RF (Supplementary Table 3). Although there is a possibility that secondary SS patients without sicca symptoms were included in the SLE and RA groups, the anti-AQP5 autoantibodies seem to be more prevalent in SS than in other autoimmune diseases.

Why do you think that AQP5 stratifies for targeted therapies? From my perspective this is a very bold statement, and I do not see a strong scientific proof of this.

→ We toned down the last sentence in Abstract and Discussion section as following: Screening anti-AQP5 autoantibodies may help to form subgroups of SS for targeted therapy.

Line 32: We do not know whether the infiltration causes dryness, autoantibodies might be involved and other things we don’t know yet (as you write later in the text).

→ “leading to” has been removed and rephrased as following: Sjögren's syndrome (SS) is a systemic autoimmune disease characterized by focal lymphocytic infiltration of the salivary/lacrimal glands and dryness of the oral cavity and eyes [1].

The association to hypergammaglobulinaemia might bias associations to other disease characteristics and immune-complex manifestations. I would add this to the limitations.

→ We apologize an error in the sentence (line 307-309) in Discussion section. The “not” was omitted by mistake, which has been corrected as following: However, the association of the high titer of anti-AQP5_E1 IgG with systemic diseases or other adverse prognostic markers, including RF, hypocomplementaemia, hypergammaglobulinemia, and the presence of germinal center was not observed. As shown in Table 5, hypergammaglobulinaemia is not associated with anti-AQP5 autoantibodies. This result was correctly described in Result section (line 224-226): Only the high FLS score was associated with the high titer of anti-AQP5_E IgG (Table 5).

Reviewer 2 Report

This manuscript reports a follow-up study to validate the prevalence of AQP-5 IgG in a non-Korean cohort. The study was well designed and the data was properly presented to support the conclusion. 

Author Response

We appreciate your time for reviewing our manuscript.

Reviewer 3 Report

Jeon et al present a case for the quantification of antiAQP-5 antibodies for stratification of pSS patients. They compare tow techniques, one employing a flourescence-based cell assay, and an ELISA. The paper is nicely written, the statistics are thoroughly done, and the authors acknowledge the major limitation of their study, the non-SS control group.

I agree with the control group not particulary conforming to control group status, considering, as stated in the discussion, that '42 of the 43 controls had a certain type of sialadentitus in the labial salivary glands'. To clarify the composition of the non-SS group, could the authors include some kind of flow chart with the breakdown of the group in terms of 'signs and symptoms' of pSS? There are additionally many medications which affect the functioning of the salivary glands (beta-blockers for example). Can the authors add information about the medication status of the non-SS group?

In terms of techniques, I read with interest the MDCK-based AQP5 assay. The technique, although apparently useful in some contexts, is unconventional. Particularly the quantification, based on 'decreasing the brightness of red signal until all signals of AQP-5 stain disappear.' For clarity in the methods, could the authors expand on exactly how they do that? If necessary, include example images of microscopy, and an anti AQP5 IgG score generated from it?

The authors also examined whether the anti-AQP5_E IgG readout could replace other criteria, in the ACR-EULAR pSS classification criteria. Can the authors explain how they arrived at a weighting of 2 for anti-AQP5_E IgG? They have removed either SSA or FLS score, both with weight of 3 from the criteria. It seems to me that they should be replaced with a new criteria of the same weighting.

Finally, in the discussion, the authors mention that  'anti AQP-5_E1 IgG was associated with low UWSFR among SICCA registry samples, but no among SS samples alone' (line 307). can the authors point out where this SS-alone data is? It's not immediately obvious to me where to look for this.

Textual: the authors appear to have left some yellow highlighted text behind (lines 217-224, 297-301 and 304-306). Please remove.

Author Response

We appreciate your time and helpful comments.

1. I agree with the control group not particularly conforming to control group status, considering, as stated in the discussion, that '42 of the 43 controls had a certain type of sialadentitis in the labial salivary glands'. To clarify the composition of the non-SS group, could the authors include some kind of flow chart with the breakdown of the group in terms of 'signs and symptoms' of pSS? There are additionally many medications which affect the functioning of the salivary glands (beta-blockers for example). Can the authors add information about the medication status of the non-SS group?

→ Unfortunately, we did not receive the data for subjective signs and symptoms and medication status from the SICCA registry. Considering the short revision period given, it was impossible to obtain new information. Although the subjective symptoms of SS do not always match with the objective test results, UWSFR, Schirmer’s test, and ocular staining listed in Table 1 provide how much percentage of non-SS group have dryness in the oral cavity and eyes. We added information on nonspecific/sclerosing chronic sialadenitis in Table 1. We also found some errors in the numbers in Table 1, which have been corrected in revision.

2. In terms of techniques, I read with interest the MDCK-based AQP5 assay. The technique, although apparently useful in some contexts, is unconventional. Particularly the quantification, based on 'decreasing the brightness of red signal until all signals of AQP-5 stain disappear.' For clarity in the methods, could the authors expand on exactly how they do that? If necessary, include example images of microscopy, and an anti AQP5 IgG score generated from it?

→ Autoantibodies to AQP4, an important biomarker for neuromyelitis optica, were discovered using AQP4-expressing mouse tissues and AQP4-overexpressing HEK293 cells (J Exp Med 2005;202:473-477), and the cell-based immunofluorescence cytochemistry is still a gold standard assay for screening the anti-AQP4 autoantibodies (reference 18). Our detail method has been published as a book chapter (reference 16), and it has been mentioned in the revised manuscript. In addition, the example images of microscopy have been included as a supplementary Figure 1.

3. The authors also examined whether the anti-AQP5_E IgG readout could replace other criteria, in the ACR-EULAR pSS classification criteria. Can the authors explain how they arrived at a weighting of 2 for anti-AQP5_E IgG? They have removed either SSA or FLS score, both with weight of 3 from the criteria. It seems to me that they should be replaced with a new criteria of the same weighting.

→ We tried both weight 2 and 3, and chose one that gave the best diagnostic power. In this particular cohort, there were no positive SSA or FLS ≥ 1, but 19 were anti-AQP5_E IgG positive in the non-SS group. Therefore, weight 3 decreased specificity a lot. 11 out of 111 SS cases did not have proper FLS score. Therefore, the replacement of FLS ≥ 1 with anti-AQP5_E IgG of weight 2 did not lose much in sensitivity. As we mentioned in Discussion section line 329 ~ 331, none of autoantibodies other than anti-AQP5 autoantibodies were detected in the non-SS SICCA registry subjects included in the current study. Whether the non-SS subjects with anti-AQP5 IgG have an increased prevalence of SS diagnosis in the future is an important question to be answered.

4. Finally, in the discussion, the authors mention that 'anti AQP-5_E1 IgG was associated with low UWSFR among SICCA registry samples, but no among SS samples alone' (line 307). can the authors point out where this SS-alone data is? It's not immediately obvious to me where to look for this.

→ It is described in the result section line 228-233.

5. Textual: the authors appear to have left some yellow highlighted text behind (lines 217-224, 297-301 and 304-306). Please remove.

→ It was the resubmission of previously submitted manuscript, and the editor asked to mark changes from the previous version. They are now removed, and new changes have been highlighted.

Reviewer 4 Report

Primary Sjögren's syndrome is perhaps one of the autoimmune diseases with the maximum delay in diagnosis due to an insidious onset, heterogeneous clinical presentations and variable course of disease progression. Unlike other autoimmune disorders, pSS lacks universally accepted classification criteria. Multiple criteria used for diag­nosis as well as classification of the disease along with nonspe­cific symptoms at presentation lead to under-diagnosis.  Today it is a great need to optimize the combination characteristics of patients with Sjögren's syndrome.

So, it is relevant to research biomarkers to stratify the complex and heterogenous phenotypes of Sjögren's syndrome.
I wonder why the researchers chose RA and SLE groups? Why not include patients with systemic sclerosis or undifferentiated connective tissue disease in this case? I would like to justify this choice in  the article.

I wonder if anti-aquaporin 5 autoantibodies have a relationship to disease activity. I would like to justify this.

Perhaps it is too bold to say that a test of anti-AQP5_E1 Ig G  can replace the FLS score. I would suggest that due to the multifaceted nature of the disease, this study of anti-aquaporin 5 autoantibodies may be needed to form subgroups of SS.

Author Response

We appreciate your time and helpful comments.

1. I wonder why the researchers chose RA and SLE groups? Why not include patients with systemic sclerosis or undifferentiated connective tissue disease in this case? I would like to justify this choice in the article.

→ In the Method section, the following sentence has been added: RA and SLE were chosen as disease controls because they are most prevalent other than SS among the connective tissue diseases, particularly in Korea [14,15].

2. I wonder if anti-aquaporin 5 autoantibodies have a relationship to disease activity. I would like to justify this.

→ Our aims of study were to validate the prevalence of anti-AQP5 IgG in a non-Korean cohort and to optimize the method for the detection of anti-AQP5 IgG. We didn’t request SSDAI or all clinical data to calculate SSDAI and had no chance to analyze the association between anti-AQP5 IgG and SSDAI. We mentioned that it needs to be clarified in Discussion section (line 338) as following: Associations between the presence of anti-AQP5 autoantibodies and the systemic manifestations of SS, in particular dryness-related symptoms and the EULAR SS disease activity index (ESSDAI), need to be clarified in future studies.  

3. Perhaps it is too bold to say that a test of anti-AQP5_E1 Ig G can replace the FLS score. I would suggest that due to the multifaceted nature of the disease, this study of anti-aquaporin 5 autoantibodies may be needed to form subgroups of SS.

→ As we described in line 303, the anti-AQP5_E1 IgG test may be used in supplementary when the labial biopsy failed to get a proper FLS score. To emphasize the complementary role, we added “Only”.

This manuscript is a resubmission of an earlier submission. The following is a list of the peer review reports and author responses from that submission.

Round 1

Reviewer 1 Report

The manuscript can be more concise.

Table 1 %(n) shows the same numbers in the (n). Authors may want to correct the numbers.

Authors stated, "in addition, strong associations of anti-AQP5 autoantibodies with the main 224 features of SS were confirmed." In stead, authors could describe what phenotype of SS is associated with anti AQP5.

The discussion of the second paragraph is lengthy and not clearly stated.

Limitation of the study was not addressed.

Author Response

We appreciate the helpful comments that improved our manuscript.

1. The manuscript can be more concise.

→ We shortened elaborative explanation in Result and Discussion sections. Although we had to add more analysis results and discussion to accommodate the other reviewers’ requests, the length of Discussion is shorter than the previous version.

2. Table 1 %(n) shows the same numbers in the (n). Authors may want to correct the numbers.

→ The (n) in Table 1 indicates the total number with proper data because some samples had missing data. To clarify it, we changed the table legend from “† Values are expressed as the percentage (total number with proper data).” to “†n present the total number with proper data.”.

3. Authors stated, "in addition, strong associations of anti-AQP5 autoantibodies with the main features of SS were confirmed." Instead, authors could describe what phenotype of SS is associated with anti AQP5.

→ It has been rewritten as following: In addition, strong associations of anti-AQP5 autoantibodies with the serologic and histopathological features of SS were confirmed.

4. The discussion of the second paragraph is lengthy and not clearly stated.

→ Most part of the second paragraph has been rewritten as following: The anti-AQP5 IgG determined by CB-IFC in the absence of epitope peptides, which was the same assay used in the previous study, resulted in a sensitivity of 0.595 and a specificity of 0.721 using the cut-off value that gives a maximal accuracy. The cut-off value to obtain a sensitivity of 0.73, the sensitivity obtained in the previous study, resulted in a specificity of 0.558. Although other reasons can be speculated, the reduced specificity can be attributed mainly to a difference in control subjects between two studies: the control subjects in the previous study were all healthy, but the non-SS controls in the current study had symptoms suggestive of SS. Importantly, 42 out of the 43 non-SS subjects had a certain type of sialadenitis in the labial salivary glands, including FLS, nonspecific chronic sialadenitis, and sclerosing chronic sialadenitis. Such inflammation in the salivary glands may contribute to the production of anti-AQP5 autoantibodies. Preincubation of sera with epitope peptides substantially decreased the signals of AQP5-staining IgG in SS samples but not in non-SS control samples. Consequently, the use of either C2 or E1 peptides in CB-IFC assays screening for anti-AQP5 IgG improved the diagnostic accuracy.

5. Limitation of the study was not addressed.

→ It has been added to the 5th paragraph of Discussion as following: This study has several limitations. First, the SLE and RA subjects did not go through the full workup for the diagnosis of SS because none of the patients fulfilled the inclusion criteria for SS. The prevalences of secondary SS in SEL and RA are 6.5%-19% and 4-31%, respectively [17]. Unlike the SICCA registry samples, the SLE and RA samples had no associations between the anti-AQP5 IgG and SSA, ANA, or RF (Supplementary Table 3). Therefore, the possibility that secondary SS was included in the SLE and RA groups is low. Second, associations of the anti-AQP5 autoantibodies with the various extraglandular and systemic manifestations of SS were not fully analyzed. Third, this was a cross-sectional study; therefore, the predictive value of the anti-AQP5 autoantibodies could not be evaluated.

Reviewer 2 Report

In this study Jeon et al. validate the utility of antibodies against aquaporin-5 (AQP5) for the diagnosis of primary Sjögren’s syndrome (SS) in Asian, non-Korean SS patients. Control groups are patients with salivary dysfunction due to non-SS sialadenitis, as well as patients with systemic lupus erythematosus (SLE) and rheumatoid arthritis (RA) without signs of SS. Furthermore, they compare two methods of detection of anti-AQP5 antibodies. As published previously from the same group, they have found elevated expression of anti-AQP5 antibodies in SS patients with anti-AQP5 IgG specific to E1 epitope to have the best discriminative capacity between SS and non-SS patients. In addition, anti-AQP5 levels were associated with other diagnostic features of SS such as SSA (and ANA) positivity, biopsy focus score≥1 and ocular staining score. Although these findings are interesting and possibly indicate a new biomarker helpful in SS diagnosis, the following issues need to be clarified.

The results of cell-based immunofluorescence (CF-IF) and ELISA assays differentiate with CF-IF being superior in SS detection and ELISA in association with other SS diagnostic features. Which method is or suggested as the gold standard? This should be clearly stated and justified. Although the discriminative ability of anti-AQP5 is not high and adequate by itself to discriminate SS patients from other patients with salivary dysfunction it could add to the other criteria and improve current diagnostic sensitivity and ability. This should also be investigated in the current study. The differences of anti-AQP5 expression between SS and non-SS control groups is due to higher expression in subgroup of SS patients. Are these the same patients in all tests? Are these a discrete patient subgroup, such as patients with systemic disease or predisposed to develop lymphoma? It could be proved that anti-AQP5 is a useful biomarker for discriminating patients at high risk for severe disease and/or lymphoma development. The correlations with other autoantibodies and positive salivary and ocular tests are not adequate to support the association of anti-AQP5 with phenotypical features of the disease. These are just other disease criteria. Associations with various clinical and serological disease features (e.g. various extraglandular and systemic manifestations, adverse prognostic markers etc.) should be examined to support such as statement.

Author Response

We appreciate the helpful comments that improved our manuscript.

1. The results of cell-based immunofluorescence (CF-IF) and ELISA assays differentiate with CF-IF being superior in SS detection and ELISA in association with other SS diagnostic features. Which method is or suggested as the gold standard? This should be clearly stated and justified.

→ CB-IFC and ELISA presented comparable diagnostic performance for SS because AUC was little bit higher for CB-IFC but the actual accuracy in the current study was reverse (AUC: 0.703 vs. 0.695, accuracy: 0.690 vs. 0.707). In the 3rd paragraph of Discussion section, we included both AUC and accuracy values to support the comparable diagnostic performance and stated that “we recommend anti-AQP5_E1 IgG ELISA as the gold standard for the diagnosis of SS” in the last sentence of the paragraph.

2. Although the discriminative ability of anti-AQP5 is not high and adequate by itself to discriminate SS patients from other patients with salivary dysfunction it could add to the other criteria and improve current diagnostic sensitivity and ability. This should also be investigated in the current study.

→ We tested whether the anti-AQP5_E IgG positivity can replace any item in the 2016 criteria, and found that the anti-AQP5_E IgG positivity with a weight of 2 could replace the FLS score ≥ 1 with quite good performance (AUC: 0.993, sensitivity: 0.973, specificity: 0.953). This result was described in Results and also discussed in the 4th paragraph of Discussion.

3. The differences of anti-AQP5 expression between SS and non-SS control groups is due to higher expression in subgroup of SS patients. Are these the same patients in all tests? Are these a discrete patient subgroup, such as patients with systemic disease or predisposed to develop lymphoma? It could be proved that anti-AQP5 is a useful biomarker for discriminating patients at high risk for severe disease and/or lymphoma development.

→ Yes, the same patients had high levels of anti-AQP5 antibodies in multiple tests. However, the multiple high titers did not have better associations with systemic disease or adverse prognostic markers. Therefore, we chose to report just high titer of anti_AQP5_E IgG (Please, see below).

4. The correlations with other autoantibodies and positive salivary and ocular tests are not adequate to support the association of anti-AQP5 with phenotypical features of the disease. These are just other disease criteria. Associations with various clinical and serological disease features (e.g. various extraglandular and systemic manifestations, adverse prognostic markers etc.) should be examined to support such as statement. 

→ We have changed “phenotypic features” in Title into “serologic and histopathological features” and that in body texts into “disease criteria”. We analyzed association of anti-AQP5_E IgG, anti_AQP5_E IgG high titer, and multiple high titers with systemic disease or several adverse prognostic markers in the 111 SS samples. Only a high FLS score (≥ 3) had an association with the anti_AQP5_E IgG high titer (Table 5). It has been described in Results section as following: We also explored whether anti-AQP5_E IgG positivity is associated with systemic diseases or adverse prognostic markers, including RF, hypocomplementemia, hypergammaglobulinemia, a high FLS score, and the presence of a germinal center [7]. Only a high FLS score was associated with a high titer of anti-AQP5_E IgG (Table 5).

Reviewer 3 Report

Please find my comments regarding the article by Jeon et al. summarizing prevalence and associations AQP5 antibodies in SjS, SLE and RA. 

You are highlighting the need for better diagnostic tests in SjS. Would be nice to see an analysis of test performance of AQP5 antibodies versus diagnostic criteria or biopsy Cannot comment on the methods 45% of RA patients were ANA positive, and SSA was only tested in 2 patients - how can you rule out secondary SjS? SSA was positive in 69% of SLE Patients, salivary and tears flow was not measured - how can you rule out secondary SjS? Was there a difference of AQP5 antibodies in SSA+ versus SSA- SLE and ANA+ versus ANA- RA`? The introduction lacks a section about the biology of AQP5 and AQP5 antibodies.  Please include the characeristics of SjS patients with focus score <1 (20%); were there any correlations between histology and AQP5 antibodies? Please report associations between IgG levels and AQP5 antibodies, moreover selective IgA deficiency should be taken into account Would be good to add a histopathologlcial staining on AQP5

Author Response

We appreciate the helpful comments that improved our manuscript.

1. You are highlighting the need for better diagnostic tests in SjS. Would be nice to see an analysis of test performance of AQP5 antibodies versus diagnostic criteria or biopsy.

→ We tested whether the anti-AQP5_E IgG positivity can replace any item in the 2016 criteria, and found that the anti-AQP5_E IgG positivity with weight 2 could replace the FLS score ≥ 1 with quite good performance (AUC: 0.993, sensitivity: 0.973, specificity: 0.953). This result was described in Results and also discussed in the 4th paragraph of Discussion.

2. Cannot comment on the methods 45% of RA patients were ANA positive, and SSA was only tested in 2 patients - how can you rule out secondary SjS? SSA was positive in 69% of SLE Patients, salivary and tears flow was not measured - how can you rule out secondary SjS? Was there a difference of AQP5 antibodies in SSA+ versus SSA- SLE and ANA+ versus ANA- RA?

→ The SLE and RA subjects did not go through the full workup for the diagnosis of SS because none of the patients fulfilled the inclusion criteria for SS. The prevalences of secondary SS in SEL and RA are 6.5%-19% and 4-31%, respectively [17]. Unlike the SICCA registry samples, the SLE and RA samples presented no association between the anti-AQP5 IgG and SSA, ANA, or RF (Supplementary Table 3). Therefore, the possibility that secondary SS was included in the SLE and RA groups is low. These have been added to the 5th paragraph of Discussion.

3. The introduction lacks a section about the biology of AQP5 and AQP5 antibodies. 

→ It has been added to the 2nd paragraph of Introduction as following: AQP5 is predominantly expressed at the apical membranes but is also expressed at the basolateral membranes of human salivary gland acini [13]. AQP5 is composed of six transmembrane helices connected with three extracellular (A, C, and E) and two intracellular (B and D) loops [13]. We previously reported the presence of anti–AQP5 immunoglobulin G (IgG) in Korean patients with primary SS with a sensitivity of 0.73 and a specificity of 0.68 [11]. We also characterized the epitopes of anti-AQP5 IgG localized to the three extracellular loops [12].

4. Please include the characeristics of SjS patients with focus score <1 (20%); were there any correlations between histology and AQP5 antibodies?

→ The SS and non-SS subjects were grouped into three groups: FLS score ≥ 1, 1 > FLS score >0, and FLS score = 0, and associations with anti-AQP5 antibodies were reanalyzed. Tables 1, 3, and 4 and Supplementary Table 1 were changed accordingly. In general, the anti-AQP5 antibodies had better associations with FLS score ≥ 1 than FLS score > 0. We also found an association of anti_AQP5_E high titer with a high FLS score (≥ 3) in the 111 SS samples and reported it (Table 5).

5. Please report associations between IgG levels and AQP5 antibodies, moreover selective IgA deficiency should be taken into account.

→ We obtained additional information on the serum IgG and IgA concentrations and association with the anti-AQP5 antibodies. None of the subject had selective IgA deficiency. The positive anti-AQP5_E IgG determined by ELISA had a week association (c2 = 4.09, P = 0.043) with hypergammaglobulinemia in the 154 SSICA registry samples, including SS and non-SS, but not in the SS samples alone. The latter one is reported in Table 5.

6. Would be good to add a histopathological staining on AQP5.

→ In order to obtain biopsy sections from the SICCA registry, the proposal has to be reviewed. It took us a year to obtain the sera samples.

Round 2

Reviewer 2 Report

The manuscript is significantly improved.

Although the analysis showed that the anti-AQP5-E IgG ELISA can replace the salivary gland biopsy in the 2016 criteria with a quite good performance, it should be clearly stated that it can only be supplementary to salivary gland biopsy since it cannot provide critical information about disease, such as the development of lymphoma or the predominating immune cell response. Furthermore, it has not been tested whether it could discriminate SS from other causes associated with sialadenitis, such as lymphoma (irrelevant to SS), sarcoidosis, IgG4-related disease etc.